ecology/environmental science/biogeography

collections, biodiversity, marine invertebrates, Mollusca, Echinodermata, Crustacea

# Natural history collections recapitulate 200 years of faunal change

Christine Ewers-Saucedo[1], Andreas Allspach[2], Christina Barilaro[3], Andreas Bick[4], Angelika Brandt[2,5], Dieter Fiege[2], Susanne Füting[6], Bernhard Hausdorf[7], Sarah Hayer[1], Martin Husemann[7], Ulrich Joger[8], Claudia Kamcke[8], Mathias Küster[9], Volker Lohrmann[10], Ines Martin[11], Peter Michalik[12], Götz-Bodo Reinicke[11], Martin Schwentner[7,13], Michael Stiller[10] and Dirk Brandis[1]

[1]Zoologisches Museum, Christian-Albrechts-Universität zu Kiel, Hegewischstraße 3, 24105 Kiel, Germany
[2]Senckenberg Research Institute and Natural History Museum, Senckenberganlage 25, 60325 Frankfurt am Main, Germany
[3]Landesmuseum Natur und Mensch Oldenburg, Damm 38-44, 26135 Oldenburg, Germany
[4]Zoological Collections of the University of Rostock, Institute for Biological Sciences, General and Systematic Zoology, Universitätsplatz 2, 18055 Rostock, Germany
[5]Goethe-University of Frankfurt, FB 15, Institute for Ecology, Evolution and Diversity, Max-von-Laue-Str. 13, 60439 Frankfurt am Main, Germany
[6]Museum für Natur und Umwelt Lübeck, Musterbahn 8, 23552 Lübeck, Germany
[7]Centrum für Naturkunde (CeNak), Martin-Luther-King-Platz 3, 20146 Hamburg, Germany
[8]Staatliches Naturhistorisches Museum, Pockelsstraße 10, 38106 Braunschweig, Germany
[9]Müritzeum, Zur Steinmole 1, 17192 Waren (Müritz), Germany
[10]Übersee-Museum Bremen, Bahnhofsplatz 13, 28195 Bremen, Germany
[11]Deutsches Meeresmuseum, Katharinenberg 14-20, 18439 Stralsund, Germany
[12]Zoologisches Museum der Universität Greifswald, Loitzer Straße 26, 17489 Greifswald, Germany
[13]Naturhistorisches Museum Wien, Burgring 7, 1140 Wien, Austria

CE-S, 0000-0003-1891-7901; BH, 0000-0002-1604-1689; VL, 0000-0003-0476-7041; PM, 0000-0003-2459-9153

**Author for correspondence:**
Christine Ewers-Saucedo
e-mail: ewers-saucedo@zoolmuseum.uni-kiel.de

Changing species assemblages represent major challenges to ecosystems around the world. Retracing these changes is limited by our knowledge of past biodiversity. Natural history collections represent archives of biodiversity and are therefore an unparalleled source to study biodiversity changes. In the present study, we tested the value of natural history collections for reconstructing changes in the abundance and presence of species over time. In total, we scrutinized 17 080

quality-checked records for 242 epibenthic invertebrate species from the North and Baltic Seas collected throughout the last 200 years. Our approaches identified eight previously reported species introductions, 10 range expansions, six of which are new to science, as well as the long-term decline of 51 marine invertebrate species. The cross-validation of our results with published accounts of endangered species and neozoa of the area confirmed the results for two of the approaches for 49 to 55% of the identified species, and contradicted our results for 9 to 10%. The results based on relative record trends were less validated. We conclude that, with the proper approaches, natural history collections are an unmatched resource for recovering early species introductions and declines.

# 1. Introduction

Since the dawn of the Anthropocene, distribution patterns and species abundances have become increasingly distorted. At an increasing frequency, species invade new habitats worldwide, shift and expand their ranges, while others decline in abundance or go extinct (e.g. [1–4]). Reconstructing changes in the species composition, however, is inherently difficult, requiring to travel back in time to compare past and present species assemblages [5]. Historical natural history collections play a major role in these efforts and are sometimes the only resource. A prominent example is the global loss of pollinating insects: a comparison of relative abundances of bumblebees from historical (1900–1999) and recent collections (2007–2009) throughout the United States provided some of the first solid evidence of their dramatic decline [6]. Other inventive examples for the use of natural history collections include the reconstruction of climate-driven geographic range retractions of Australian seaweeds and American mangroves [7,8], of phenological changes in Tibetan plants [9], and of emerging mismatches between flowering time and pollinator emergence [10].

These studies emphasize the value of natural history collections as a resource for detecting temporal changes in geographic range, phenology and abundance. More specifically, they have a much longer record than ecological long-term surveys, which did not take off until the 1980s, when the first evidence of far-reaching changes in our ecosystems emerged. In addition, natural history collections represent a physical record of past species occurrences [11,12], allowing re-identification of specimens and re-examination of specific characters, such as size or reproductive activity long after the specimens were collected [13]. Moreover, they can be used to study the specimens' parasites, microbiome, toxicology and food sources via stable isotope signatures [14].

The use of natural history collections to assess changes over time is challenging given the often unsystematic nature of the collecting efforts driven by individual interests of researchers and institutions alike: collectors may have focused on certain taxa instead of collecting all species present, and sampled different locations throughout time [15,16]. Even if collectors re-sampled consistently throughout their career, such long-term efforts may not be continued after the collectors retire. Not all of the collected specimens have been preserved and deposited in natural history collections, a necessary consequence of the cost of maintaining collections [17]. This may be especially true for common species. They are probably less often represented in collections than actually present in nature. Furthermore, historical collections are often sparse due to the loss and destruction of collections during the two world wars. Figure 1 summarizes the common steps from the collecting of specimens to the natural history record database.

Changes in marine ecosystems are less directly observable than changes in terrestrial ecosystems. Consequently, most evidence for changes in marine biodiversity to date comes from economically important species with extensive fishery records, such as cod, oysters and lobsters, which turned from cheap street food to expensive seafood between the nineteenth and twentieth century [18–21]. Two of the few marine regions with long-term data are the North and Baltic Seas, which underwent substantial community shifts in the late 1980s due to increasing seawater temperatures [22–28]. These shifts are exemplified by the replacement of cod by jellyfish and horse mackerel. The decline of this important top predator probably increased the abundance of its prey, such as decapods and echinoderms, while in turn their prey, bivalves, declined in abundance. This top-down regime shift was first uncovered by analysing North Sea plankton sampled from 1946 on and was confirmed by comparing dredged benthos samples from different decades [29–31].

For the present study, we analysed the information provided by natural history collection records to reconstruct changes in the relative abundance and presence of marine invertebrates of the North and

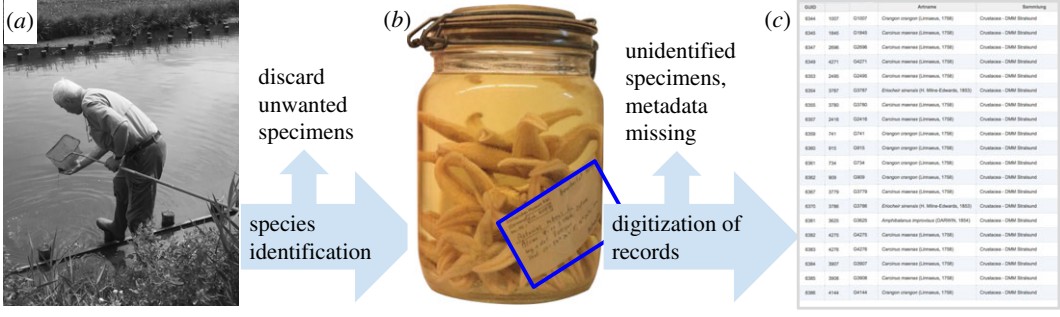

**Figure 1.** Workflow that led to the investigated collection database. (*a*) Sampling of specimens, for example by hand-sampling in coastal areas, or by dredging aboard research vessels. (*b*) Deposition of specimens that are preserved and identified ideally to the species level. If and which specimens are discarded depends on the sampling design, the individual researchers' preferences and the available space in the collections. Collection records reflect the fact that all specimens of the same species and sampling event are stored together as a single lot. (*c*) Digitized collection records allow the analysis of century-old natural history collections.

Baltic Seas, and validate the results by comparison with independent sources. In order to reconstruct long-term changes in the marine invertebrate communities in these regions, we compiled a database of records from 12 German natural history museums and university collections with extensive marine invertebrate samples dating back to the nineteenth century. Including many smaller collections is necessary because Germany does not have a national natural history museum, which means that type specimens, first records of certain species and extensive cruise samples are distributed across large and small natural history collections. We tested different statistical approaches to detect species with changes in abundance over time, in particular species that have become rare or gone extinct, which we call declining species, and species that have increased in abundance. This category holds species that have expanded their range naturally into the North or Baltic Sea from other, mostly more southern parts of European waters, as well as invasive species, which have been introduced by humans from other parts of the world and species that have increased in abundance.

# 2. Material and methods

## 2.1. Database preparation

We aimed to investigate temporal changes in the marine invertebrate assemblages of the North and Baltic Seas, two marine regions heavily impacted by humans. This study focuses on three abundant marine invertebrate taxa: Crustacea, Echinodermata and Mollusca. The basis of our analyses is the collection records associated with specimens stored in natural history collections, in particular the location and time of collection, and the species identification. We relied on previously digitized collection records (e.g. [32]) and newly digitized a considerable amount of data for the relevant collections of the Zoological Museum Kiel (ZMK), the Center of Natural History Hamburg (ZMH), the Senckenberg Research Institute and Natural History Museum (SMF), as well as collections of several natural history collections in northern Germany belonging to a network of museums from the North and Baltic Seas region called NORe (http://nore-museen.de/) (electronic supplementary material, S1). The resultant 'Aquila' database is hosted by SMF, accessible at https://marsamm.senckenberg.de.

## 2.2. Data curation

We attempted to provide geographic coordinates for each record for subsequent data interpretation. While geographic coordinates were not available for all collection records, most of them came with a description of the respective sampling location. These were first matched to the descriptions of a georeferenced locality present in the investigated database. Subsequently, we used the function 'geocode' of the 'ggmap' package [33] in the R environment [34] to georeference all remaining localities. When these automatic approaches failed to find an accurate match, we attempted to manually match localities by interrogating the database and the Internet for the core information contained in the locality descriptions. Subsequently, we excluded records of collections outside of the

North Sea and German Baltic Sea using the spatial analysis R package 'sp' [35,36]. Regions with few records were not included in the analysis.

Several natural history records had outdated species designations. We validated species names for all records using the World Register of Marine Species ('WoRMS', [37]). Given a large number of investigated specimens, we checked the species identification only when the validation suggested discrepancies.

We minimized the problem of misidentifications by focusing on taxa for which a majority of deposited specimens were identified to species level. For Echinodermata, these are Asteroidea (sea stars), Ophiuroidea (brittle stars) and Echinoidea (sea urchins). For Mollusca, we focused on Gastropoda (snails), Bivalvia (bivalves) and Polyplacophora (chitons). For Crustacea, we focused on Thoracica (barnacles) and Decapoda (shrimps, crabs and lobsters).

To alleviate issues related to different sampling techniques through space and time, we only included taxa that were sampled by hand and dredging, which is most of the epibenthic macrofauna. We removed infauna, i.e. burrowing bivalves, as well as species with a mean size of 1 cm or smaller from the analyses. These two groups were not adequately represented in the digitized historical collections. Despite these measures, biases may remain in natural history collections.

One bias that may distort our perception of changing species abundance is that not all dredged or captured specimens are deposited in the natural history collections, but only a few voucher specimens, as during some of the large-scale cruises that deposited their material in SMF. This may be especially true for common species. We attempted to address this issue by considering the number of instances a species was deposited instead of the number of individuals that were deposited in each lot.

## 2.3. Database validation

The subsequent analyses assume that the investigated natural history collection records reflect natural abundances, i.e. that more common species have been deposited more often and have consequently more collection records than rare species. To test this assumption broadly, we first compared species numbers with known species lists for North and Baltic Seas [38]. To obtain a more detailed comparison, we used the relative abundances provided by the German Red List for benthic marine invertebrates [39]. The Red List assesses the current status and long-term changes in biodiversity predominantly based on the results of governmental surveys. For all species present in both the investigated database and the Red List, we compared the number of records with the categorical abundance criterion for each species of the Red List, e.g. 'extinct', 'rare', 'relatively common', and tested for significant differences between categories with a generalized linear model with a Poisson error distribution (function 'glm' of the 'stats' package in R). If the investigated collection database reflects natural abundances, more common species should have more records. We do not expect a perfect fit as the relative abundances of the Red List reflect the current status of those species, whereas the investigated database integrates collections over the course of two centuries.

## 2.4. Identifying faunal changes

To identify faunal changes from the investigated database, we applied three related approaches. The first two approaches split the data into two time periods and compared the distribution and frequency of each species' records between the historical and more recent time period. We set the time split at the end of the first multi-year set of cruises into the North and Baltic Seas, the year 1912. We assume that at that point in time all the common, and some rare species, were collected and deposited in the natural history collections. To identify potential neozoa, we focus on species deposited in the collections after 1912. To identify declining species, we use the fact that more recent collections contain many more specimens and lots, and that any species collected less frequently than the other species in more recent times may have declined in abundance. The third approach considers relative increases and declines in the annual number of records over time for each species. The records for each species are set in relation to higher, interspecific taxa, which may alleviate issues with different sampling intensity over the years.

### 2.4.1. Potential neozoa

To identify species that have established themselves in the North and Baltic Seas during the last 150 years, we followed the reasoning that a species collected and deposited only in recent times may be an addition to the region, either as an alien introduced species, or as a species that expanded its range

into the region. We define 'recent times' as any species collected after the end of the 'Terminfahrten' (after 1912). We excluded species that were only collected in a single year, had fewer than 10 records or were only collected between 1977 and 1990, during the most extensive collecting efforts of the research vessel RV *Senckenberg* cruises. These species are probably overall rare, so that their presence in collections is rather due to increased sampling effort than to true changes in their abundance or geographic range.

Species that fit the selection criteria were searched for publications in Google Scholar identifying these species as invasive or range-expanding using the species name and the terms 'invasive', 'non-native' or 'range expansion'. We also searched in neozoan databases for the North Sea (www.nobanis.org, https://easin.jrc.ec.europa.eu/easin), published neozoan species lists [40] and evaluated by independent occurrence data from the Ocean Biogeographic Information System (OBIS, www.iobis.org) and the Global Biodiversity Information Facility (GBIF, www.gbif.org). OBIS and GBIF host geographic information from various sources, e.g. museum collections, literature and citizen science observations (https://www.gbif.org/what-is-gbif). These resources are not completely independent of the investigated database, as previously digitized collections are present in the GBIF database, and the OBIS database contains at least some of the 'Terminfahrten' records, compiled by Stein *et al.* [32] and analysed in previous studies [30,31]. To avoid this issue, we excluded these records in the online databases prior to our assessments. We also evaluated relative species trends with GBIF's 'Relative observation trends' tool (https://www.gbif.org/tool/1IVlBHIXeUK68ac6okOqkm/relative-observation-trends). This tool calculates the annual proportion of observations for a species in relation to the total number of observations for a higher taxonomic unit the species belongs to. It then tests for significance with a linear regression model. To further ensure that these species were not collected from historically undersampled regions, we mapped collecting locations in different time periods.

### 2.4.2. Potentially declining species

The aforementioned analysis did not identify declining species. When filtering for species that were only deposited prior to 1991 or earlier, we found that most species were from geographic regions not sampled well after 1991, such as the Kattegat, the northern North Sea and British coasts. Instead, we took advantage of the uneven temporal sampling. Declining species should have been collected more in the past than in recent times. Given the increased sampling efforts since the 1980s, any species that was collected more often during the 'Terminfahrten' (1902–1912) and before than in subsequent years is a clear candidate for decline. For each species, we assessed whether the proportion of collection records after 1912 for each species is significantly less than the overall proportion of all records deposited after 1912 using the function 'pbinom' in the R package 'stats' [34].

We mapped past and recent collecting locations for all species with relatively few records after 1912 to ensure that these species were not collected from undersampled locations. Furthermore, the mapping allowed us to assess whether their relatively low recent rate of deposition in the investigated natural history collections was linked to changes in their geographic distribution. We validated our findings by comparing the species we identified as potentially declining with the species listed as endangered in the Red List for benthic marine invertebrates [39]. As expected, the majority of records in the GBIF and OBIS databases were from the last 50 years, which did not allow a straightforward assessment of decline in a species' abundance. Instead, we used GBIF's 'Relative observation trends' tool described under 'Potential neozoa'.

### 2.4.3. Relative record trends

We also calculated changes in the relative abundance of species over time to estimate faunal changes. This approach is implemented in GBIF's 'Relative observation trends' tool, which we adopted for the investigated database in the R environment [34]. We used both the class and the phylum as the higher taxonomic units and tested for significant linear regression trends over time with the function 'lm' in the 'stats' package [34]. We consider trends as significant for when the relative number of records changed significantly with regard to either phylum or class, or both. We excluded species that increased relative to the respective class, but declined compared with the phylum, or vice versa. We excluded years with fewer than five records for the higher taxonomic units, and species with fewer than 10 records overall. The focused sampling and deposition of specimens from the Dogger Bank beginning in the 1990s could mimic an increase in the abundance of species that are common at the Dogger Bank. We therefore repeated the analysis excluding records of specimens from the Dogger

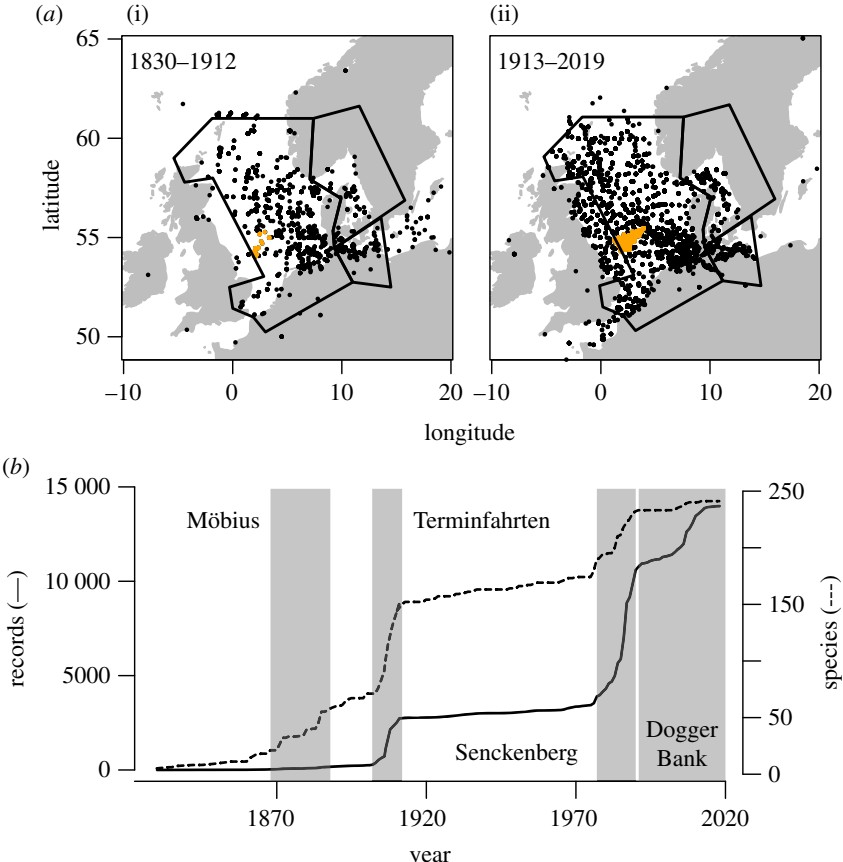

**Figure 2.** Spatio-temporal distribution of investigated collection records. (*a*) Spatial distribution of records before and after 1912. The polygons denote the investigated area, notably the North Sea without the British coast, the German part of the Baltic Sea and the transition area between the two seas. (*b*) Accumulative number of deposited collection lots and species that were new to the collections. Grey areas highlight the major collecting efforts detailed under 'Data curation' in the Results section, responsible for the largest increases in species and collection lots.

Bank, and only consider changes in abundance significant if they are recovered in the full dataset as well as the dataset without Dogger Bank records.

# 3. Results

## 3.1. Data curation

Initially, the database contained 30 692 records. Of those, 4983 records were not georeferenced but contained a description of the sampling locality. These localities were first matched to the georeferenced localities within the investigated database. This identified 89 identical localities. The function 'geocode' of the 'ggmap' package allowed us to georeference 4535 of the remaining localities. This left us with 359 localities where automatic approaches failed to find an accurate match; of those, 250 could not be matched, and 109 were improperly matched. These localities were matched by manual web searches. For 214 records, the sampling locality was either not specified or could not be mapped, and these records were excluded from subsequent analyses, as were 101 records that lay outside our specified North Sea–Baltic Sea area (figure 2*a*).

In the next step, we removed 1625 records that did not have the collection date specified. After filtering out taxa that were not well-represented, we retained 18 480 records. Another 279 lots were not identified to the species level, and were removed from subsequent analyses. The British coast and eastern Baltic Sea were poorly represented in the collections in historical and recent times, respectively (figure 2*a*), and 1055 records originating from those two regions were removed. Of the remaining 17 080 records in the final dataset, over half were Crustacea (9781 records), followed by Echinodermata with 4092 records

(24%) and Mollusca with 3207 records (19%). A total of 242 species were investigated in the database: 48% (115 spp.) belonged to Crustacea, 15% (36 spp.) to Echinodermata and 38% (91 spp.) to Mollusca.

## 3.2. Database validation

A comparison with the species listed for the German North and Baltic Seas [38] shows a good match between species numbers. Of the crustacean species in the final dataset, 100 belong to Decapoda and 15 to Thoracica. Zettler *et al*. [38] report 88 Decapoda and 10 Thoracica, indicating that the collections contain more species than reported for the German North and Baltic Seas. The collections' database contains 36 echinoderm species, while Zettler *et al*. report 40 species. Lastly, the database contains 91 molluscan species, 21 Bivalvia, 66 Gastropoda and four Polyplacophora, while Zettler *et al*. [38] report over 100 species each for the Bivalvia and Gastropoda, i.e. 143 Bivalvia and 218 Gastropoda. The relatively low number of molluscan species and records is due to the exclusion of infauna and species of small size in our analyses and should not be taken as a general lack of Molluscan specimens in natural history collections.

Specimens were unevenly collected and deposited both in time and space, corresponding to known sampling efforts (figure 2*b*). The oldest specimens in our records belong to *Cancer pagurus* Linnaeus, 1758, collected near Helgoland in 1830 and deposited at the SMF (catalogue no. SMF Cr 2931). Additional individual specimens from the nineteenth century are deposited in the Staatliches Naturhistorisches Museum Braunschweig (SNMB), the Zoologisches Museum der Universität Greifswald (ZIMG), the Müritzeum Waren (MUW), the Übersee Museum Bremen (UMB) and the Landesmuseum Natur und Mensch Oldenburg (LMNM).

The first systematic cruises and collections in the North and Baltic Seas took place between 1868 and 1888, when Karl August Möbius was professor and director of the zoological collections at the University of Kiel [41–44]. More extensive cruises in the North and Baltic Seas took place between 1902 and 1912. These cruises, known as the 'Terminfahrten', formed the first large-scale, multi-annual effort to systematically collect the fauna of the North and Baltic Seas [32,45–48]. The respective collections are housed at the ZMK.

Few new specimens were deposited to the investigated collections between 1913 and 1976, but records from the Museum für Natur und Umwelt Lübeck (MNUL), the Müritzeum Waren (MUW) and the Centrum für Naturkunde (CeNak) der Universität Hamburg (ZMH) provide temporal continuity in the data. From 1977 to 1990, SMF undertook extensive sampling efforts of North Sea fauna. Since 1991, SMF has conducted annual sampling cruises to the Dogger Bank, a shallow bank in the central North Sea. This relatively warm area is considered a 'hotspot' for warm water species. The Baltic Sea fauna was assessed by smaller scale efforts of the Deutsches Meeresmuseum Stralsund (DMM), the Zoologische Sammlungen Rostock (ZSRO) and ZMK.

Half of the investigated species (118 spp.) were present both in the investigated collection database and the German Red List for benthic marine invertebrates [39]. The remaining 123 species of the investigated database have not yet been assessed by the German Red List. The recent relative abundance categories of the Red List significantly explained the frequencies of the investigated database ($p$-value < 0.005). Moreover, rare species had fewer records than common species (figure 3), suggesting that the number of investigated collection records reflects natural abundances broadly. As expected, there was significant overlap between categories: some species with more than 500 records in the investigated database were categorized as rare, such as *Hyas coarctatus* Leach, 1815. *Crangon allmanni* Kinahan, 1860 had the largest number of collection records with 1131 records and was listed as 'relatively abundant'. By contrast, the blue mussel *Mytilus edulis* was listed as 'very abundant', but had 'only' 400 records. The following analyses may resolve whether these discrepancies are due to biases in the collections or the result of considerable real changes in the abundance of these outlier species within the last 150 years.

## 3.3. Identifying faunal changes

### 3.3.1. Potential neozoa

We excluded 37 species from the analysis that were only recorded during the intense sampling efforts of the RV *Senckenberg* cruises in the 1980s (electronic supplementary material, S4), 43 species with less than 10 records and 29 species collected and deposited only in a single year (electronic supplementary material, S2). Of the remaining species, 19 were only deposited in the investigated collections after

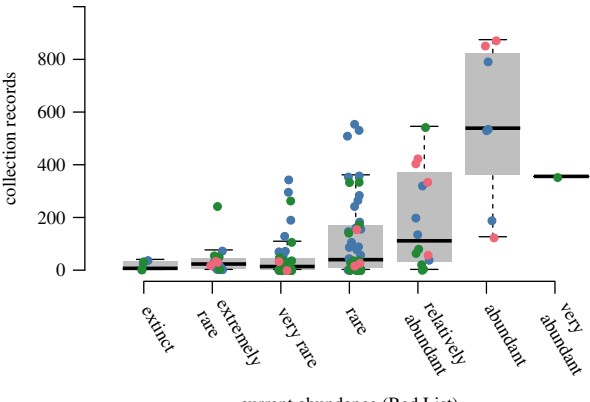

**Figure 3.** Relationship between the relative abundances as reported in the German Red List for benthic marine invertebrates [39] and the number of collection records for these species in the investigated collection database. Relative abundances are sorted from least abundant (extinct) to most abundant (very abundant). Colours denote supraspecific taxa: Crustacea (blue), Echinodermata (red) and Mollusca (green).

1912 (figure 4) and will be analysed in more detail. The late appearance of the snail *Steromphala cineraria* (Linnaeus, 1758) and the spider crab *Inachus phalangium* (Fabricius, 1775) in the investigated database are probably the result of uneven spatio-temporal sampling and deposition (see electronic supplementary material, S5 for details).

Besides these two sampling artefacts, this approach identified several species that have previously been identified as neozoa to the North Sea. Eight species are known as invasive alien species ([40], www.nobanis.org): the bay barnacle *Amphibalanus improvisus* (Darwin, 1854), the Harris mud crab *Rhithropanopeus harrisii* (Gould, 1841), the Asian shrimp *Palaemon macrodactylus* Rathbun, 1902, the Chinese mitten crab *Eriocheir sinensis* H. Milne Edwards, 1853, the Asian shore crabs *Hemigrapsus sanguineus* (De Haan, 1835) and *H. takanoi* Asakura & Watanabe, 2005, the American slipper limpet *Crepidula fornicata* (Linneaus, 1758), and the Pacific oyster *Magallana gigas* (Thunberg, 1793).

Another four species are confirmed warm water species that have expanded their ranges into the North Sea from more southern latitudes during the last 30 years: *Goneplax rhomboides* (Linnaeus, 1758) [49,50], *Necora puber* (Linnaeus, 1767) [51–53], *Thia scutellata* (Fabricius, 1793) [49,54] and *Diogenes pugilator* (P. Roux, 1829) [55]. It is important to note that most of these species were identified as range-expanding based on a subset of the data we analysed, specifically based on the collections of the SMF from the 1970s on.

In addition to these known range expansions, we identified six species that may have undergone range expansions into the central North Sea or increases in abundance in the last century, but have not been reported as such (figure 4): the three-spined shrimp *Philocheras trispinosus* (Hailstone in Hailstone & Westwood, 1835), the estuarine shrimp *Palaemon longirostris* H. Milne Edwards, 1837, the crab *Ebalia tuberosa* (Pennant, 1777), the swimming crab *Liocarcinus marmoreus* (Leach, 1814), the hairy hermit crab *Pagurus cuanensis* Bell, 1845, and the spider crab *Inachus phalangium* (Fabricius, 1775). They were not collected during the Terminfahrten (1902–1912). The spatio-temporal spread apparent from the investigated database, which we describe briefly for each species in the electronic supplemental material, S5, matches generally the observation records in GBIF and OBIS and early species accounts.

Several recent neozoa for the North Sea listed by Tsiamis *et al.* [40] were not present in the collection database. These species have been reported from the North or Baltic Sea since the 2000s [56–58] and may have been expected in the collection database. In particular, these are five warm water species: the Chinese hat snail *Calyptraea chinensis* (Linnaeus, 1758), the European abalone *Haliotis tuberculata* Linnaeus, 1758, the scallop *Pecten maximus* (Linnaeus, 1758), the marbled crab *Pachygrapsus marmoratus* (Fabricius, 1787) and the topshell *Phorcus lineatus* (da Costa, 1778). In addition, five recent invasive species were not present in the collections: the shrimp *Penaeus japonicus* Bate, 1888, the mud crab *Dyspanopeus sayi* (Smith, 1869), the predatory snail *Rapana venosa* (Valenciennes, 1846), as well as the oyster drills *Ocinebrellus inornatus* (Récluz, 1851) and *Urosalpinx cinerea* (Say, 1822).

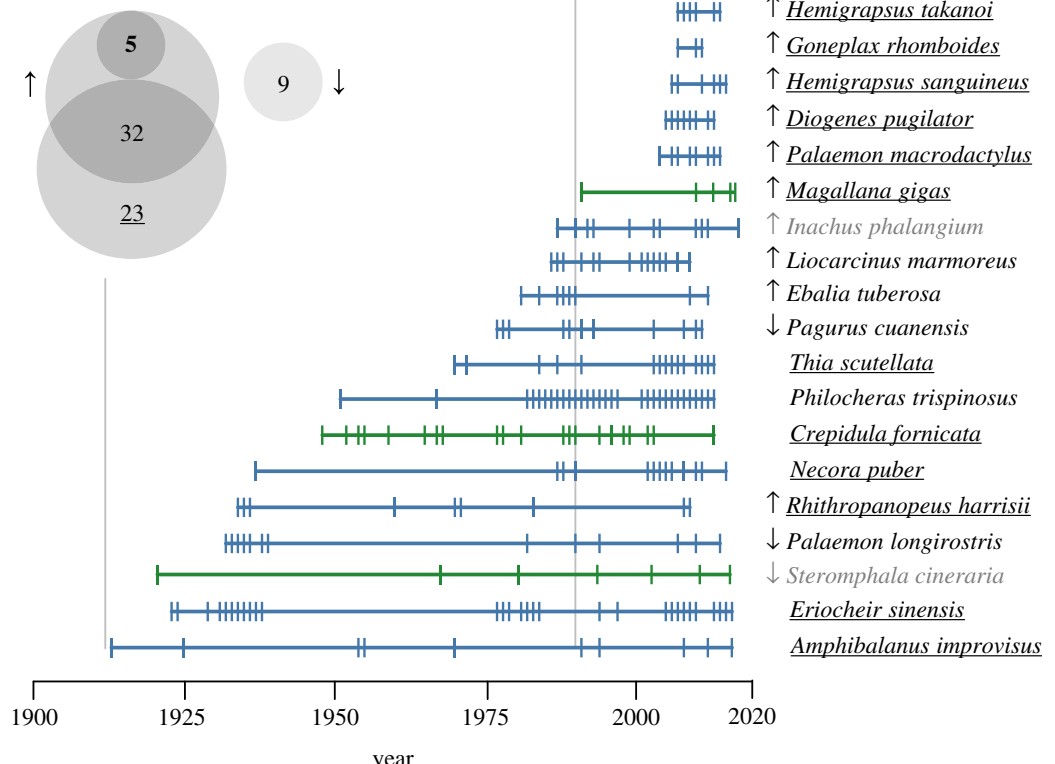

**Figure 4.** Temporal distribution of collections records for potential neozoa. Temporal distribution of records for species appearing after 1912 in the investigated collection database, excluding species identified as rare. The years 1912 and 1991 (the end of two intense collecting efforts) are indicated by grey vertical lines. Underlined species names indicate previously known neozoa for the North or Baltic Sea. Names of known endangered species [39] are bolded, and names of spatio-temporal sampling artefacts are shown in grey. Arrows indicate species that were significantly increasing or declining based on the GBIF Relative Observations Tool. Colours denote supraspecific taxa: Crustacea (blue) and Mollusca (green). Our analysis did not identify any neozoan Echinodermata to the North or Baltic Sea. The Venn diagram in the upper corner summarizes the cross-validation results from the GBIF Relative Observations Tool (arrows), German Red List (bold number) and known neozoa (underlined number) as per cent species.

In summary, we identified 17 species, representing 7% of all investigated species, as neozoa of the North Sea's fauna. Of those, eight species are known invasive species that were introduced into the North Sea by humans from a different part of the world, i.e. Asia and North America. Four species have been previously identified as warm water range expansions, entering the North Sea in the wake of its warming, while another five species may represent to-date unnoticed range expansions into the area. These results are mostly concordant with GBIF observation trends: of the species appearing only after 1912 in the collections' database, 10 had positive GBIF observation trends, while three species had negative observation trends, one of which we identified as a spatio-temporal artefact. The individual species patterns invite specific studies to evaluate their current status and distribution ranges.

### 3.3.2. Potentially declining species

As expected, most species were collected more often after 1912 than before (electronic supplementary material, S2 and S6). The proportion of all lots deposited after 1912 was 83.2%. Using this overall percentage as our baseline, we identified a total of 49 species that were deposited less frequently than expected (21% of all investigated species).

The species we identified as declining included 18 species considered 'endangered' or 'extinct' by the German Red List (figure 5, electronic supplementary material, S2). We also identified the brittle star *Ophiothrix fragilis* (Abildgaard in O.F. Müller, 1789) as declining, which is listed as 'potentially endangered' by the German Red List. GBIF's Relative Observation Trends Tool found congruent

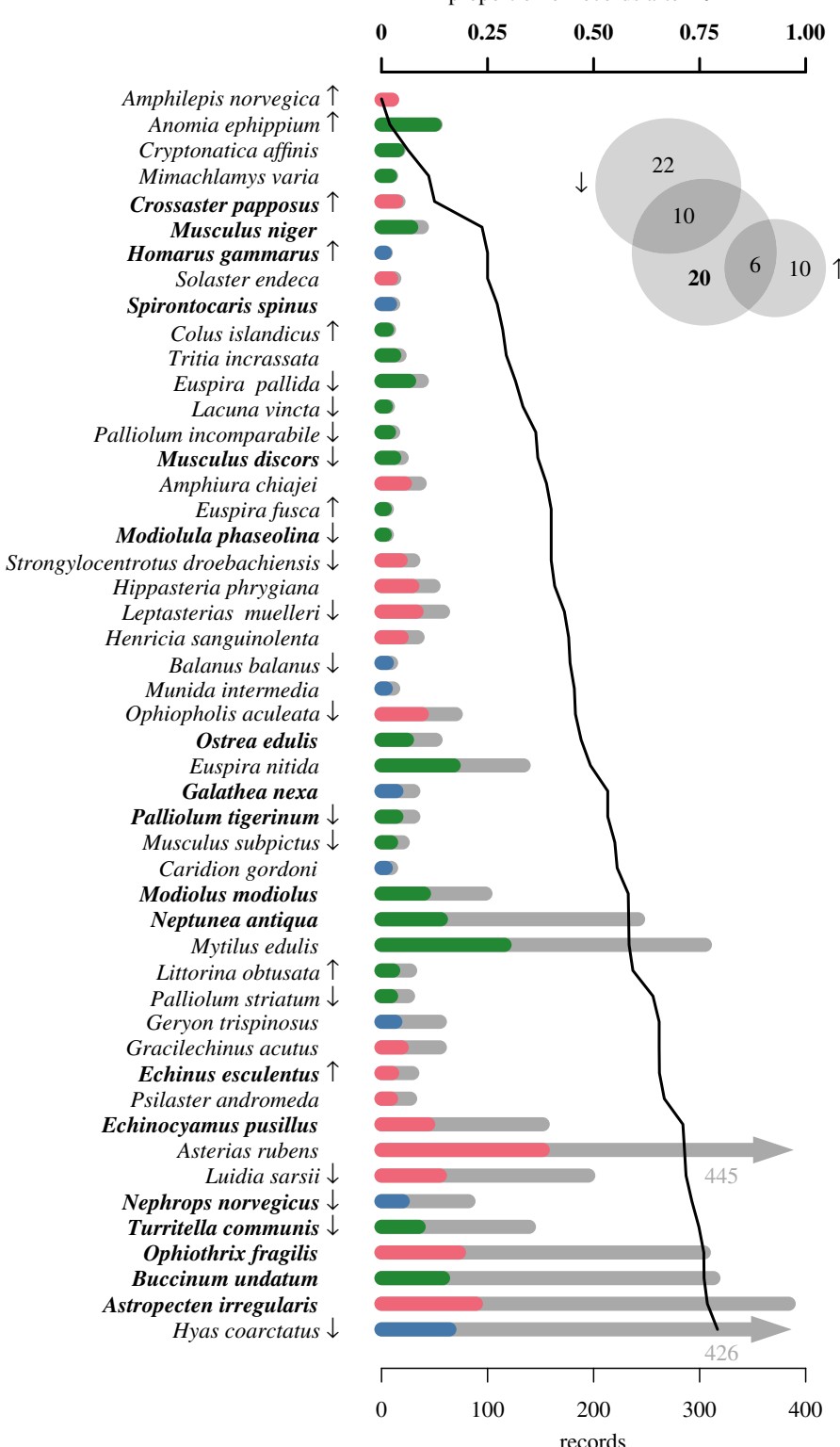

**Figure 5.** Potentially declining species with relatively few records after 1912. The proportion of records after 1912 is shown by the bold line (top axis). Bar plot of the number of records available before and after 1912 as coloured and grey bars, respectively. Colours denote the supraspecific taxon: Crustacea (blue), Echinodermata (red) and Mollusca (green). Names of known endangered species [39] are bolded. Arrows indicate species that were significantly increasing or declining based on the GBIF Relative Observations Tool. The Venn diagram in the upper corner summarizes the cross-validation results from the GBIF Relative Observations Tool (arrows) and German Red List (bold number) as per cent species.

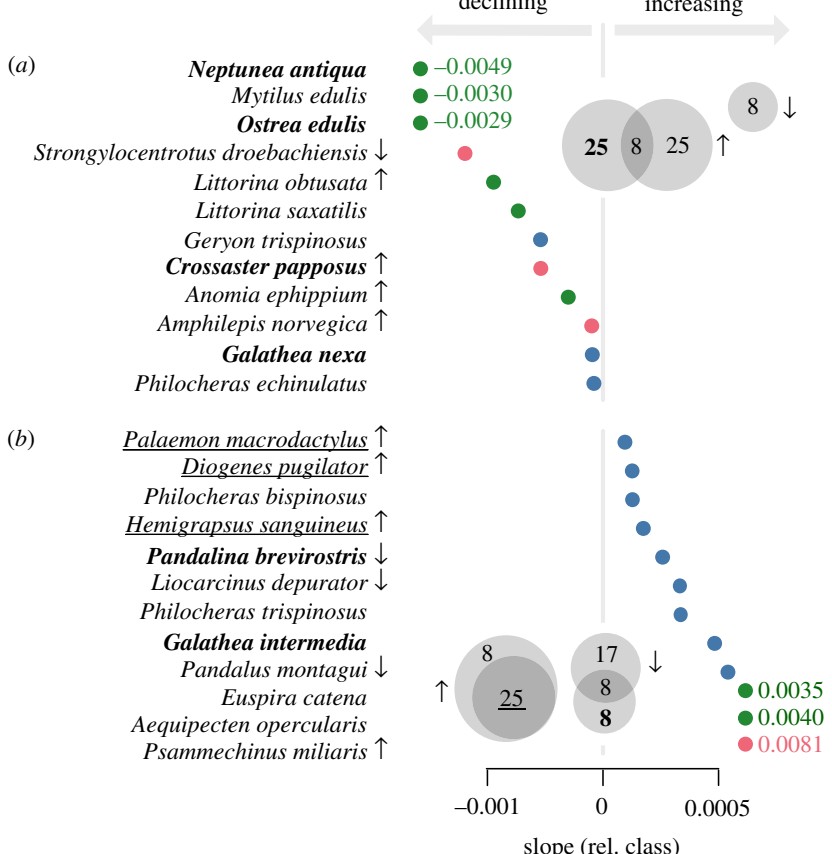

**Figure 6.** Regression slope values of species with significant record trends. (*a*) Species with significantly negative slopes, (*b*) species with significantly positive slopes. Depicted is the slope in relation to the class for the complete dataset. Colours denote intraspecific taxa: Crustacea (blue), Echinodermata (red) and Mollusca (green). Names of known endangered species [39] are bolded. Underlined species are independently identified neozoa to the North or Baltic Sea (see text for references). Arrows indicate species that were significantly increasing or declining based on the GBIF Relative Observations Tool. The Venn diagrams in each corner summarize the cross-validation results from the GBIF Relative Observations Tool (arrows), German Red List (bold percentage) and known neozoa (underlined percentage) as per cent of species.

results for 16 species; their relative number of observations declined over time (figure 5, electronic supplementary material, S2). Increases in the proportion of GBIF observations, contradicting our findings, were documented for eight species (figure 5). Curiously, most of these species had few records in the investigated database.

### 3.3.3. Relative record trends

Relative record trends were significantly negative or positive for 12 species each (figure 6, electronic supplementary material, S2). Four of the species with negative trends were considered endangered by the German Red List [59], and two species had negative GBIF observation trends (electronic supplementary material, S2, figure 6). Contradicting our inferences were four species with positive GBIF observations trends, one of which was considered endangered. Of the 12 species declining in relative records, 10 species were also identified as potentially declining by the previous approach (figure 7). Only the snail *Littorina saxatilis* (Olivi, 1792) and the shrimp *Philocheras echinulatus* (M. Sars, 1862) were identified as declining by the relative records tool.

   All species identified as potentially increasing in abundance occur throughout the North Sea (electronic supplementary material, S3), which means that uneven spatio-temporal sampling is unlikely to be responsible for the observed relative increase in records for these species. In line with our findings, four species have positive GBIF observation trends (electronic supplementary material, S2, figure 6). Contradicting evidence was present for two species that are considered endangered by the German Red List [59], and three species that are declining in relative GBIF observations over time (electronic

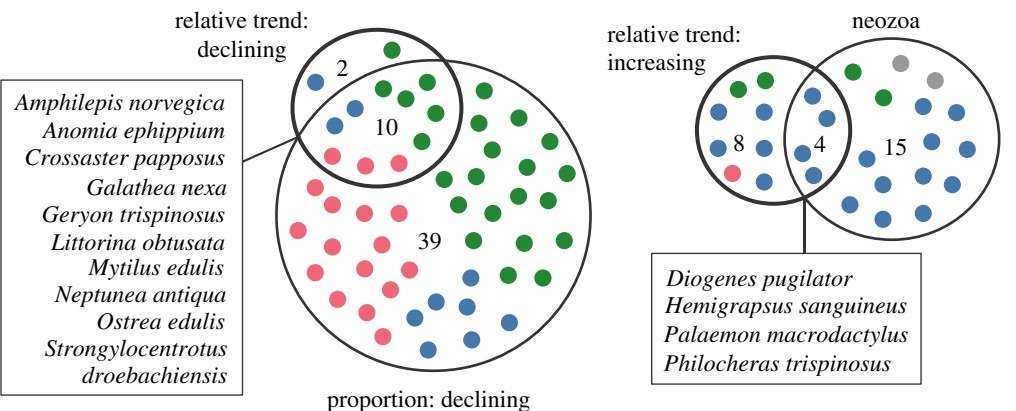

**Figure 7.** Intersecting results between approaches. Each filled circle represents one species. The colours denote intraspecific taxa: Crustacea (blue), Echinodermata (red) and Mollusca (green).

supplementary material, S2, figure 6). For four species, neither corroborating nor contradicting evidence from independent sources was available.

# 4. Discussion

Natural history collections have recently been recognized as rich resources to identify trait changes and range shifts in response to global change [6–8,10]. Their potential to reconstruct changes in species' abundances is less well understood. In the present study, we applied straightforward statistical approaches to identify neozoa as well as species declining or increasing in abundance over time and compared our results with independent data sources. This cross-validation indicates good concordance for two of the employed approaches. In the following, we will discuss the usefulness and limitations of natural history collections to reconstruct biodiversity changes, as well as the appropriateness of our approaches and cross-validation sources. Lastly, we suggest future research avenues and discuss broad-scale implications.

## 4.1. Limitations of natural history collections

Natural history collections represent an assembly of cruise samples, taxon-specific collections, voucher specimens for taxonomic and biogeographic studies, and collections acquired from other museums or private collectors ([15,16], present study). In consequence, any attempt to identify faunal changes from natural history collections is potentially compromised by uneven sampling in both space and time. In the investigated natural history collections, spatially uneven sampling and deposition of specimens was apparent at the borders of the North Sea, i.e. the Kattegat in the East, the British coast in the West, the Shetland Islands in the North and the English Channel in the South. These regions were not represented sufficiently at all times, at least partially caused by the changing geopolitical landscape that emerged during the two world wars. This resulted in some species only occurring at specific times, erroneously suggesting their recent appearance or decline. To avoid false inferences, we excluded regions with few records during specific time periods, such as much of the British coast. We did so after conducting the analyses including this region and realizing that a number of species were spatio-temporal artefacts. Second, we were able to identify species as spatio-temporal artefacts by mapping their collecting locations against the localities of the collecting sites in the investigated database. Lastly, we conducted some analyses with and without a certain region—the Dogger Bank in our case—to understand the impact of uneven sampling on our inferences. Including additional collections with different geographic foci will further alleviate these issues.

Temporally uneven sampling is exemplified in the investigated database by the large sampling efforts from the 1980s on. In the 1980s, many previously unrecorded species appeared in the investigated database. At first glance, this could be interpreted as large faunal changes. It is, however, more probable that these newly recorded species are actually rare and were collected and deposited only in the 1980s because of the increased collecting efforts at that time. This conclusion is supported by the fact that these newly recorded species have not been deposited in the collections of the investigated

database before or after, which allowed us to classify them as rare species rather than neozoa. Indeed, appropriate replication enhances the quality of the data and can partly overcome the constraints of sampling with particular gear [60].

Spatio-temporal unevenness is not only caused by uneven sampling and deposition efforts. Wars in general, and World War II in particular, have caused catastrophic losses in the investigated collections and collections across Europe. This further imbalances the proportion of historical to more recent records. The shrimp *Palaemon serratus* (Pennant, 1777), for example, a rocky habitat specialist, has a restricted distribution in the North and Baltic Seas, occurring mainly around Helgoland. Pre-World War II records for this region were predominantly deposited at the ZMH, but destroyed during World War II. The loss of pre-World War II records could mimic a recent increase in the species' records. Unfortunately, this issue is difficult to account for. We consulted species accounts from the beginning of the twentieth century and independent collection records available at OBIS and GBIF to understand if species were indeed rare or absent prior to the world wars. In general, the inclusion of collections less impacted by wars may alleviate these issues.

## 4.2. Advantages of natural history collections

While the analysis of natural history collections to reconstruct faunal change is not without challenges, collections have unparalleled advantages. Our analyses initially identified some species as changing in abundance that were artefacts of changing taxonomy. The neozoan approach, for example, initially identified three species, the squat lobster *Galathea dispersa* Bate, 1859, the shrimp *Processa modica* Williamson in Williamson & Rochanaburanon, 1979 and the snail *Epitonium clathrus* (Linnaeus, 1758). These species have cryptic morphologies and complex taxonomic histories [42,48,61,62]. We hypothesized that this may have caused an initial lack of records for those species prior to 1912. Consequently, we re-identified the deposited specimens of the genera in question based on current identification keys. In all three cases, the presence of these three species in the North Sea as early as 1872 was validated. We are certain that such taxonomic issues are common, but can be solved with museum collections—and only with natural history collections. Literature reviews or ecological survey data do not allow the re-identification of investigated specimens. This makes museum collections invaluable for long-term comparisons across generations of taxonomists.

The second advantage of natural history collections is that they provide the temporal continuity to distinguish between rare visitors to an area and established populations. More generally, by integrating records across centuries, they allow us to distinguish between cyclical changes in abundance or those caused by natural variation, and long-term trends. Callaway *et al*. [30], for example, compared survey data from three time periods, the data of the Terminfahrten (1902 to 1912), data from the English groundfish surveys (1982 to 1985), and trawling data from a biodiversity monitoring project of the European Union (2000). By comparing only three time periods, they could not distinguish between problems with data collection and real changes in abundance.

### 4.2.1. How appropriate were our approaches?

In general, we consider our approaches to be conservative. We focused on avoiding 'false positives', which means that we probably removed several species because their data were insufficient, even though they actually changed in abundance. Many more species than we were able to identify have probably undergone changes. Incorporating additional natural history collections will be a valuable step towards avoiding these issues and validating our results.

For the first two approaches aimed at identifying neozoa and declining species, respectively, we compared two large time windows, and integrated data across decades. These approaches should be 'blind' to short-term changes in abundance, but at the cost of resolution. This means we are unable to pinpoint when species changed in abundance exactly. This coarseness is a necessary consequence of the amount of data needed to arrive at solid conclusions and the data distribution across time.

Neozoa in general, either invading or expanding species, may be particularly suited to be detected in natural history collections, because they are rather striking at the time of discovery and more likely to be collected and deposited in collections. In addition, we also identified some species as potential neozoa that had not previously been identified as such, suggesting that the long-term records of natural history collections could unveil additional species that are not native to an area, or have increased substantially in abundance. The fact that a number of recent neozoa were not present in the investigated collections, on the other hand, stresses the importance of continuing collecting efforts,

both large scale and punctual, and the deposition of specimens into natural history collections. Natural history collections maintain specimens sustainably over centuries, as evident from the present and many other studies [6,7,9,11,14,16,17].

The two approaches identifying declining species overlapped broadly in their results, albeit the trends approach identified far fewer species. A large overlap between these two approaches is expected because both are based on the number of records over time. Conversely, only four species with relative increases in record numbers were also identified as potential neozoa (figure 7). This relatively low concordance between the two approaches is a consequence of the different analytical approaches: one approach considers only species occurring relatively late in the investigated databases (after 1912), but not changes in the number of records, which is the criterion for the relative records approach. In particular, the observation trend estimation considers also species present prior to 1912, as long as the species increased in its relative proportion of records over time.

The relative records trend approach in general may be problematic because species of the same higher taxonomic unit are not statistically independent from each other. For example, if the abundance of one species increases or declines, the relative abundance of other species of this taxon may decline or increase, respectively, even if they did not actually change in abundance. This causes a particular bias for supraspecific taxa with few species. Ideally, enough species are in a supraspecific taxon to provide sufficient 'background noise', against which each species can be evaluated. On the other hand, if too many species are in the supraspecific taxon, the slope for each single species will be very shallow and may not be significant. We dealt with this issue by comparing the number of records of each species against two taxonomic ranks, the class and the phylum.

In line with our concerns for the relative records trend approach, this approach was much less cross-validated than the two approaches comparing two large time windows. While we attempted to account for the much higher deposition of specimens since the 1980s, this may still have caused some aberrant pattern. It is also possible that some of the cross-validation sources do not provide accurate results.

## 4.3. How valid was our cross-validation?

We faced the challenge to find independent, reliable sources to cross-validate our results. We used multifarious sources, i.e. known species lists for declining species [39], invasive species [40] and species that have expanded their range into the North Sea [49,51,52,54,55], early species accounts of the area [45–48,63,64], as well as the large public geographic information databases OBIS and GBIF.

The cross-validation sources were not always consistent with each other, nor with our results. The two approaches comparing two large time windows had a high validation success: supporting evidence was found for 49 to 55% of the species, and less than 10% of the species identified as neozoan or declining had contradicting evidence. The independent sources contradicted each other in a few cases amounting to 0 to 8% of the identified species. The most inconsistent results were attained with the relative records trend approach. Only one-third of the identified species were validated by external sources, one-third did not have independent evidence for change, and about one-third of the species showed contradicting results.

Incongruent results could be caused by different spatio-temporal foci of the different datasets, as well as difficult or changing taxonomy. The German Red List, for example, considers only the German parts of the North and Baltic Seas, whereas the GBIF and OBIS databases include most of the North Sea. This spatial disparity caused, for example, contradicting results for the European lobster *Homarus gammarus*. While its population has strongly declined in the German part of the North Sea, where it is restricted to the waters around Helgoland, it is abundant in the remaining range with slowly increasing landings [65–67]. Thus, our analyses, which were only based on specimens from Helgoland, and the German Red List indicate a decline of this species, while the GBIF Relative Observations Trend Tool indicated a significant increase in records.

GBIF and OBIS contain both more recent results of ecological studies, very recent citizen science efforts as well as natural history collection records, and in general much more data than the investigated database. Therefore they should provide more detailed results. Nonetheless, some species with the lowest proportion of records after 1912 did not have any OBIS or GBIF records until the 1920s for the area we investigated, such as the brittle star *Amphilepis norvegica* (Ljungman, 1865) and the common jingle shell *Anomia ephippium* Linnaeus, 1758. This lack of early observations probably caused these species to have positive GBIF observation trends (figure 5). The reason for this discrepancy between GBIF and our results is a consequence of the high quality and age of the 'Terminfahrten' collections. The brittle star *Am. norvegica*, for example, was described from

Scandinavian waters, off the Swedish coast [68]. Thus this species was found by others in the area, and a direct search of the Gothenburg Museum of Natural History online collections database provides a record of *Am. norvegica* from 1865 from the Skagerrak region, and an undated record determined by Ljungman—possibly the type specimen (catalogue no. Echinodermata: 41, 42). These records are not integrated into the GBIF database, unlike a large fraction of Gothenburg's records. An integration of the investigated database with GBIF, which is underway, will provide important temporal resolution for these species. This highlights the importance of the investigated collections in particular, and of the digitization and analyses of smaller collections in general.

Noteworthy are also the shell collections of the European oyster *Ostrea edulis* Linnaeus, 1758. *Ostrea edulis* went locally extinct in the Wadden Sea and much of the North Sea in the 1950s [21,69], but did not have a significantly negative GBIF observations trend. We suggest that this confirms the persistence of oyster shells in the environment. We, on the other hand, identified it as declining with both approaches, even though shells of *O. edulis* were deposited in the investigated collections until 2000.

### 4.3.1. How accurate are our results?

Two pieces of evidence indicate that the investigated collections reflect natural species assemblages. On the one hand, the number of species present in the investigated collections matches the number of species reported in recent species lists for the area [38], suggesting that the collections are reasonably complete. On the other hand, the current abundance categories of the German Red List [39] were significantly correlated with the number of records per species. This reflects that the number of records may be an indicator of the actual abundance of a species.

Assuming therefore that our results reflect realistic faunal changes, we would like to highlight some large-scale taxonomic patterns. The majority of neozoa we identified belong to the Crustacea, a few to the Mollusca, but none to the Echinodermata. Similarly, the majority of species with positive observation trends are Crustacea, few Mollusca and only one Echinodermata, the sea urchin *Psammechinus miliaris* (P.L.S. Müller, 1771) (figure 7). The lack of invasive Echinodermata appears to be quite common: about 150 marine invasive species in Europe are Crustacea and 150 species are Mollusca, whereas only 12 species are Echinodermata, none of which are invasive to the North Sea [40]. One possible reason for the low number of Molluscan species is probably the amount of records in the investigated database (19%). This would also explain why so many Crustacea were identified; they dominate the investigated database with 57% of all records.

However, the potentially declining species exhibit the opposite taxonomic pattern: 46% and 33% of all declining species belong to the Mollusca and Echinodermata, respectively, while only 19% of the declining species are Crustacea (figure 7). This suggests that taxonomically uneven data availability is not responsible for the taxonomic biases we observed. Consequently, it implies broad-scale taxonomic shifts, some of which have been reported previously. In particular, an increase of decapod crustaceans and a decline in bivalves has been linked to top-down predation effects ultimately caused by increasing temperatures and declining cod stocks in the late 1980s [22–24,28].

The actual species identified as increasing in abundance since the 1980s by long-term ecological studies included crabs *Carcinus maenas* (Linnaeus, 1758) and *Liocarcinus holsatus* (Fabricius, 1798), the shrimps *Pandalus montagui* Leach, 1814 [in Leach, 1813–1815] and *Crangon allmanni* Kinahan, 1860 [27,70]. Conversely, the brittle star *Ophiura albida* Forbes, 1839 increased in abundance after cold winters in the 1980s, and declined in subsequent years [25,27]. The molluscs *Buccinum undatum* Linnaeus, 1758 and *Colus gracilis* (da Costa, 1778), the hermit crabs *Anapagurus laevis* (Bell, 1845 [in Bell, 1844–1853]), *Pagurus prideaux* Leach, 1815 [in Leach, 1815–1875] and *P. pubescens* Krøyer, 1838 as well as the spider crab *Hyas coarctatus* declined in abundance [26]. Increasing temperatures were repeatedly inferred as the cause of these faunistic changes [25–27,70]. Of these species, we only identified the spider crab *H. coarctatus* as declining. Overall, these studies identified different species than we did, probably as a consequence of their narrower geographic and temporal scale, as well as statistical differences. For these studies, species present in many years and at many stations increase the statistical power to discern changes in abundance. Rare species, on the other hand, have an inherently small sample size and therewith limited statistical power.

Our analysis of museum collections focuses on long-term and large-scale changes and is geared towards identifying species that do not appear in all temporal horizons. In this regard, ecological long-term studies and the analyses of museum collection metadata can complement each other. Moreover, ecological long-term studies are more valuable at discerning the causal agents of change due to their fine-scale temporal sampling, which allows correlations to environmental variance. The

investigated museum collections lack the temporal resolution to correlate changes in species composition to environmental variables. Including a larger number of collections from museums around the North Sea would not only increase the temporal resolution, but also allow us to fill the spatial gaps caused by geopolitical shifts. A good congruence between our museum metadata analysis and long-term ecological studies exists for warm water species that expanded into the North Sea recently, such as the crab *Goneplax rhomboides* (e.g. [52,53]).

The relatively large number of declining echinoderm species (17 species) contradicts previous studies based on plankton samples, which recorded an increase in echinoderm larvae from 1949 onward [23,24,28]. This discrepancy may be a result of our focus on numbers of species that changed, but not overall numbers of records per phylum. The plankton survey, on the other hand, which provided the first ecosystem-wide evidence for the large-scale community shift, evaluated changes in the abundance of echinoderm larvae, but did not identify the actual species these larvae belonged to [23,24,28]. Certain echinoderm species, such as *Psammechinus miliaris*, may have increased more in abundance than others decreased, leading to an overall increase in Echinodermata, but a decrease in actual species diversity.

Besides these broad-scale taxonomic shifts, some genera appear to be particularly prone to decline. All represented members of the mussel genus *Musculus*, the predatory snail genus *Euspira* and the pecten genus *Palliolum* had relatively few recent records, suggesting their decline in abundance. The mussel genus *Musculus* may not have been impacted strongly by fisheries [71], which was implied for other bivalves [31], and may explain the decline of the pecten genus *Palliolum*. For all three genera, the increase in temperature could have caused their decline in the North Sea, as this is the most dominant factor across a wide range of taxa [22–28]. These genera should be closely monitored, as the decline of complete genera could have significant impacts on ecosystem functioning.

### 4.3.2. How can we further confirm faunal changes inferred from natural history collections?

The analyses of other collection databases that cover a similar geographic and temporal range would be a very promising avenue, for example, a thorough analysis of the GBIF and OBIS databases. We did not attempt such an analysis at present because, as we have shown, the included natural history collections need to be understood well and potential taxonomic issues solved with the collections at hand to account for biases in the collections. This should be the goal of a large team of experts from the incorporated collections.

Geographic shifts in a species' range, which are often implied to occur during climate change, should be visible in the collection data [72]. Range expansions or retractions may be reconstructed from collection records by comparing the geographic range at different time periods [8]. This approach requires a sufficient amount of records to reconstruct the range during each time period reasonably well [72]. These assumptions could not be met with certainty using the investigated collection database. Conducting the analyses on the species for which we had the most records did not reveal any range shifts (data not shown). We conclude that this type of analysis may be successful using more data from a larger geographic range.

A different approach uses the preserved specimens itself in museum genomic analyses [73,74]. Both decline and expansion should leave population genetic signatures, i.e. changes in effective population size [75–80], which can be inferred from a few specimens [81]. This is certainly an exciting new route to explore the potential of natural history collections.

## 5. Conclusion

Natural history collections are sustainable, century-long repositories of biodiversity. Moreover, they represent the unique opportunity to re-identify specimens of uncertain species affinity and are therewith a unique source to reconstruct long-term biodiversity trends if natural history collections reflect natural species assemblages. In the present study, we showed that species abundance and collection record abundance are indeed correlated and may be used to reconstruct faunal changes over time. This means that natural history collections should not only be probed to identify trait and range shifts, but they also provide vital information on species' abundances, which can aid in developing a much-needed baseline of the historical biodiversity.

Data accessibility. The Aquila database containing the raw natural history collection records is accessible at https://marsamm.senckenberg.de.

Authors' contributions. D.B. conceptualized the study. C.E.-S. analysed the data and wrote a first draft of the manuscript. All authors generated data by digitizing collection records, and substantially revised the manuscript. All authors read and approved the final manuscript.

Competing interests. We declare we have no competing interests.

Funding. We would like to thank the German 'Bundesministerium für Bildung und Forschung' for their financial support by funding the 'MARSAMM' project grant no. (01UQ1711A).

Acknowledgements. We would like to thank the German 'Bundesministerium für Bildung und Forschung' for their financial support by funding the 'MARSAMM' project (01UQ1711A). We are also grateful for the help of the dedicated students Nele Heuer, Mona Kühn, Melina Kurzawe, Rebekka Leßke, Anna Mai, Zoe Moesges, Jakob Pfefferle and Linda Schmitz, who digitized collections at the ZMK and SMF. The MARSAMM dataportal was skilfully designed and developed by Viorel Cazan, Thomas Hörnchenmeyer, Jan Reitz, Alexander Schmid and Thomas Winter (all SMF). Two anonymous referees made valuable comments on the first version of the manuscript.

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
