## [Peer Review File · Royal Society Open Science]

Review History

RSOS-201983.R0 (Original submission)

Review form: Reviewer 1 (Ingrid Kröncke)

Is the manuscript scientifically sound in its present form?

Yes

Are the interpretations and conclusions justified by the results?

Yes

Is the language acceptable?

Yes

Do you have any ethical concerns with this paper?

No

Have you any concerns about statistical analyses in this paper?

No

Recommendation?

Accept with minor revision (please list in comments)

Comments to the Author(s)

RSOS-201983: Ewers-Saucedo, C. et al. Natural history collections reconstruct 200 years of faunal change

The paper presents the results of a very interesting approach to use natural history collection data to reconstruct faunal change. The paper reveals the importance of natural history collections as archives of biodiversity changes. The authors found trends in species changes especially for larger and dominant species similar to earlier studies for collection data.

The paper is well written and structured and the authors used comprehensive statistical analyses for the analysis of their data.

They discussed the pro and cons of using collection data, e.g. increasing sampling effort and correctness of species identification over time. Indeed, collection data can be used especially regarding presence/absence analysis of species as well as presence and increase of invasive species

In contrast, the attempt to use collection data for reconstructing changes in species abundance over time might be a problem, because collections data especially of larger epifauna species might only represent a subsample of the total catch.

I am missing in the discussion the comparison of the collection data with data from long-term ecological studies, such as the papers given below. Senckenberg am Meer is studying the epifauna communities in the south-eastern North Sea since 1998 on annual basis as part of Senckenberg's LTER North Sea Benthos Observatory. Similar studies are continued in the Dutch EEZ by colleagues from NIOZ and in the UK EEZ by colleagues from CEFAS. The comparison with the ecological data will provide the possibility to compare and confirm trends in abundance in collection data at least for the last two to three decades.

The discussion is in some parts repeating own results, which should be shortened and more focused to references. Being an ecologist, I am missing the discussion of reasons for the faunal change, which might not be the major focus of this paper, but will arise automatically while comparing trends with those from ecological studies.

I recommend the paper for publication after major revision.

Relevant papers to be considered:

Callaway, R., Alsvag, J., de Boois, I., Cotter, J., Ford, A., Hinz, H., Kröncke, I., Lancaster, J., Piet, G. Prince, P., Ehrich, S. (2002). Diversity and community structure of epibenthic invertebrates and fish in the North Sea. *ICES J. Sea. Res.* 59: 1199-1214.

Meyer, J., Kröncke, I., Bartholomä, A., Dippner, J.W., Schückel, U. (2016). Long-term changes in species composition of demersal fish and epifauna species in the Jade area (German Wadden Sea/ North Sea) since 1972. *Estuar. Coast. Shelf Sci.* 181: 284-293.

Neumann, H., Ehrich, S., Kröncke, I. (2008). Spatial variability of epifaunal communities in the North Sea in relation to sampling effort. *Helgol. Mar. Res.* 62: 215-225.

Neumann, H., Ehrich, S., Kröncke, I. (2008). Temporal variability of an epibenthic community in the German Bight affected by cold winter and climate. *Clim. Res.* 37:241-251.

Neumann, H., Ehrich, S., Kröncke, I. (2009). Variability of epifauna and temperature in the northern North Sea. *Mar. Biol.* 156:1817-1826.

Neumann, H., Reiss, H., Rakers, S., Ehrich, S., Kröncke, I. (2009). Temporal variability of southern North Sea epifauna communities after the cold winter 1995/1996. *ICES J. Mar. Sci.* 66: 2233-2243.

Neumann, H., de Boois, I., Kröncke, I., Reiss, H. (2013). Climate change facilitated range expansion of the non-native Angular crab *Goneplax rhomboides* into the North Sea. *Mar. Ecol. Prog. Ser.* 484: 143-153.

- Neumann, H., Diekmann, R., Emeis, K.-C., Kleeberg, U., Moll, A., Kröncke, I. (2017). Full-coverage spatial distribution of epibenthic communities in the south-eastern North Sea in relation to habitat characteristics and fishing effort. *Marine Environmental Research* 130: 1-11.
- Rees, H. L., Pendle, M. A., Waldock, R., Limpenny, D. S., and Boyd, S. E. 1999. A comparison of benthic biodiversity in the North Sea, English Channel, and Celtic Seas. – *ICES Journal of Marine Science*, 56: 228–246
- Reiss, H., Kröncke, I. (2004). Seasonal variability of epibenthic communities in different areas of the southern North Sea. *ICES J. Mar. Sci.* 61(6): 882-905.
- Sonnewald, M., Türkay, M., 2012a. The megaepifauna of the Dogger Bank (North Sea): Species composition and faunal characteristics 1991-2008. *Helgoland Marine Research* 66(1), 63–75.
- Sonnewald, M., Türkay, M., 2012b. Environmental influence on the bottom and near-bottom megafauna communities of the Dogger Bank: A long-term survey. *Helgoland Marine Research* 66(4), 503–511.
- Sonnewald, M., Türkay, M., 2012c. Abundance analyses of mega-epibenthic species on the Dogger Bank (North Sea): Diurnal rhythms and short-term effects caused by repeated trawling, observed at a permanent station. *Journal of Sea Research* 73: 1-6.
- Zühlke, R., Alsvag, J., de Boois, I., Cotter, J., Ehrich, S., Ford, A., Hinz, H., Jarre-Teichmann, A., Jennings, S., Kröncke, I., Lancaster, J., Piet, G., Prince, P. (2001). Epibenthic diversity in the North Sea. *Senckenbergiana marit.* 31(2): 269-281.
- Etc.

Review form: Reviewer 2 (Jan Weslawski)

Is the manuscript scientifically sound in its present form?

Yes

Are the interpretations and conclusions justified by the results?

Yes

Is the language acceptable?

Yes

Do you have any ethical concerns with this paper?

No

Have you any concerns about statistical analyses in this paper?

No

Recommendation?

Accept with minor revision (please list in comments)

Comments to the Author(s)

I have a great difficulty with this paper. This is very interesting and important contribution to science and shall be published. On the other hand it is a typical product of the „big data science“ that can be produced about books in libraries, or minerals in geological museums. While this very paper is about living organisms. Yet, there is no natural history here – no information on the environmental context, climate, habitats or life traits of examined species.

Authors put great effort to present in most objective and bias free way the issue of museal collections, still it is a description of collection of certain class of objects from a politically designated area (German Baltic is treated differently prior to II WW and before). If some species were collected in 1900 near Konigsberg in Eastern Prussia there is no reason to ignore its

presence in 1980 Kaliningrad in Soviet Union – while the only problem is that recent one is not in the German museum.

The analysis are very convincing, statistical treatment looks great – I am just curious what kind of results authors will get introducing the same method for the paleontological species – the same 200 years of history – will it show increase in abundance, distribution shifts and decline of some fossil species within 200 years time ?

Still this paper may indicate some important phenomena, that needs to be checked – habitat wise, climate and industrial stress considered etc...

I would recommend the paper to be published with a title changed to something like :

„do changes in natural history collections reflect the changes in nature ? „

With some sentence in the intro – or conclusion that authors are not presenting the natural phenomenon – rather indicate that it may exist.

Decision letter (RSOS-201983.R0)

Dear Dr Ewers-Saucedo

On behalf of the Editors, we are pleased to inform you that your Manuscript RSOS-201983 "Natural history collections reconstruct 200 years of faunal change" has been accepted for publication in Royal Society Open Science subject to minor revision in accordance with the referees' reports. Please find the referees' comments along with any feedback from the Editors below my signature.

Please submit your revised manuscript and required files (see below) no later than 7 days from today's (ie 02-Mar-2021) date. Note: the ScholarOne system will 'lock' if submission of the revision is attempted 7 or more days after the deadline. If you do not think you will be able to meet this deadline please contact the editorial office immediately.

on behalf of Dr Joachim Mergeay (Associate Editor) and Pete Smith (Subject Editor)
openscience@royalsociety.org

Associate Editor Comments to Author (Dr Joachim Mergeay):

Comments to the Author:

Dear Dr Ewers-Saucedo,

We have finally received two reviews of your paper. Both reviewers appreciate the approach used, and agree that natural history collections use in this way are valuable archives to document species turnover.

Both reviewers also highlight the need for at least some discussion on the causes of the changes in the faunal assemblages over time, and a comparison with other data sources, such as data collected in long-term ecological studies. Reviewer 1 provides some suggestions.

The second reviewer also points out that a change in a country's territory may also limit the use of natural history collections (see discussion).

Sincerely,

Reviewer comments to Author:

Reviewer: 1

Comments to the Author(s)

RSOS-201983: Ewers-Saucedo, C. et al. Natural history collections reconstruct 200 years of faunal change

The paper presents the results of a very interesting approach to use natural history collection data to reconstruct faunal change. The paper reveals the importance of natural history collections as archives of biodiversity changes. The authors found trends in species changes especially for larger and dominant species similar to earlier studies for collection data.

The paper is well written and structured and the authors used comprehensive statistical analyses for the analysis of their data.

They discussed the pro and cons of using collection data, e.g. increasing sampling effort and correctness of species identification over time. Indeed, collection data can be used especially regarding presence/absence analysis of species as well as presence and increase of invasive species

In contrast, the attempt to use collection data for reconstructing changes in species abundance over time might be a problem, because collections data especially of larger epifauna species might only represent a subsample of the total catch.

I am missing in the discussion the comparison of the collection data with data from long-term ecological studies, such as the papers given below. Senckenberg am Meer is studying the epifauna communities in the south-eastern North Sea since 1998 on annual basis as part of Senckenberg's LTER North Sea Benthos Observatory. Similar studies are continued in the Dutch EEZ by colleagues from NIOZ and in the UK EEZ by colleagues from CEFAS. The comparison with the ecological data will provide the possibility to compare and confirm trends in abundance in collection data at least for the last two to three decades.

The discussion is in some parts repeating own results, which should be shortened and more focused to references. Being an ecologist, I am missing the discussion of reasons for the faunal change, which might not be the major focus of this paper, but will arise automatically while comparing trends with those from ecological studies.

I recommend the paper for publication after major revision.

Relevant papers to be considered:

Callaway, R., Alsvag, J., de Boois, I., Cotter, J., Ford, A., Hinz, H., Kröncke, I., Lancaster, J., Piet, G., Prince, P., Ehrich, S. (2002). Diversity and community structure of epibenthic invertebrates and fish in the North Sea. *ICES J. Sea. Res.* 59: 1199-1214.

Meyer, J., Kröncke, I., Bartholomä, A., Dippner, J.W., Schückel, U. (2016). Long-term changes in species composition of demersal fish and epifauna species in the Jade area (German Wadden Sea/ North Sea) since 1972. *Estuar. Coast. Shelf Sci.* 181: 284-293.

Neumann, H., Ehrich, S., Kröncke, I. (2008). Spatial variability of epifaunal communities in the North Sea in relation to sampling effort. *Helgol. Mar. Res.* 62: 215-225.

Neumann, H., Ehrich, S., Kröncke, I. (2008). Temporal variability of an epibenthic community in the German Bight affected by cold winter and climate. *Clim. Res.* 37:241-251.

Neumann, H., Ehrich, S., Kröncke, I. (2009). Variability of epifauna and temperature in the northern North Sea. *Mar. Biol.* 156:1817–1826.

Neumann, H., Reiss, H., Rakers, S., Ehrich, S., Kröncke, I. (2009). Temporal variability of southern North Sea epifauna communities after the cold winter 1995/1996. *ICES J. Mar. Sci.* 66: 2233-2243.

Neumann, H., de Boois, I., Kröncke, I., Reiss, H. (2013). Climate change facilitated range expansion of the non-native Angular crab *Goneplax rhomboides* into the North Sea. *Mar. Ecol. Prog. Ser.* 484: 143–153.

Neumann, H., Diekmann, R., Emeis, K.-C., Kleeberg, U., Moll, A., Kröncke, I. (2017). Full-coverage spatial distribution of epibenthic communities in the south-eastern North Sea in relation to habitat characteristics and fishing effort. *Marine Environmental Research* 130: 1-11.

Rees, H. L., Pendle, M. A., Waldock, R., Limpenny, D. S., and Boyd, S. E. 1999. A comparison of benthic biodiversity in the North Sea, English Channel, and Celtic Seas. – *ICES Journal of Marine Science*, 56: 228–246

Reiss, H., Kröncke, I. (2004). Seasonal variability of epibenthic communities in different areas of the southern North Sea. *ICES J. Mar. Sci.* 61(6): 882-905.

Sonnewald, M., Türkay, M., 2012a. The megaepifauna of the Dogger Bank (North Sea): Species composition and faunal characteristics 1991-2008. *Helgoland Marine Research* 66(1), 63–75.

Sonnewald, M., Türkay, M., 2012b. Environmental influence on the bottom and near-bottom megafauna communities of the Dogger Bank: A long-term survey. *Helgoland Marine Research* 66(4), 503–511.

Sonnewald, M., Türkay, M., 2012c. Abundance analyses of mega-epibenthic species on the Dogger Bank (North Sea): Diurnal rhythms and short-term effects caused by repeated trawling, observed at a permanent station. *Journal of Sea Research* 73: 1-6.

Zühlke, R., Alsvag, J., de Boois, I., Cotter, J., Ehrich, S., Ford, A., Hinz, H., Jarre-Teichmann, A., Jennings, S., Kröncke, I., Lancaster, J., Piet, G., Prince, P. (2001). Epibenthic diversity in the North Sea. *Senckenbergiana marit.* 31(2): 269-281.

Etc.

Reviewer: 2

Comments to the Author(s)

I have a great difficulty with this paper. This is very interesting and important contribution to science and shall be published. On the other hand it is a typical product of the „big data science“ that can be produced about books in libraries, or minerals in geological museums. While this very paper is about living organisms. Yet, there is no natural history here – no information on the environmental context, climate, habitats or life traits of examined species.

Authors put great effort to present in most objective and bias free way the issue of museal collections, still it is a description of collection of certain class of objects from a politically designated area (German Baltic is treated differently prior to II WW and before). If some species were collected in 1900 near Königsberg in Eastern Prussia there is no reason to ignore its

presence in 1980 Kaliningrad in Soviet Union – while the only problem is that recent one is not in the German museum.

The analysis are very convincing, statistical treatment looks great – I am just curious what kind of results authors will get introducing the same method for the paleontological species – the same 200 years of history – will it show increase in abundance, distribution shifts and decline of some fossil species within 200 years time ?

Still this paper may indicate some important phenomena, that needs to be checked – habitat wise, climate and industrial stress considered etc...

I would recommend the paper to be published with a title changed to something like :

„do changes in natural history collections reflect the changes in nature ? „

With some sentence in the intro – or conclusion that authors are not presenting the natural phenomenon – rather indicate that it may exist.

===PREPARING YOUR MANUSCRIPT===

===PREPARING YOUR REVISION IN SCHOLARONE===

Author's Response to Decision Letter for (RSOS-201983.R0)

See Appendix A.

Decision letter (RSOS-201983.R1)

Dear Dr Ewers-Saucedo,

It is a pleasure to accept your manuscript entitled "Natural history collections recapitulate 200 years of faunal change" in its current form for publication in Royal Society Open Science.

on behalf of Dr Joachim Mergeay (Associate Editor) and Pete Smith (Subject Editor)
openscience@royalsociety.org

Associate Editor Comments to Author (Dr Joachim Mergeay):
Associate Editor
Comments to the Author:
Dear Dr Ewers-Saucedo,

Thank you for your timely revision. I feel like you have sufficiently addressed the questions raised by the reviewers. Well done!

Sincerely,
Joachim Mergeay

Appendix A

Dear reviewers and editors,

We appreciate your valuable feedback and suggestions. Below are our point-by-point responses to your concerns (bolded). Please also see the MS (tracked changes) for the exact wording we used.

Let us know if we did not address your concerns sufficiently.

Sincerely,

Christine Ewers-Saucedo

Point-by-point response

Associate Editor Comments to Author (Dr Joachim Mergeay):

Comments to the Author:

Dear Dr Ewers-Saucedo,

We have finally received two reviews of your paper. Both reviewers appreciate the approach used, and agree that natural history collections use in this way are valuable archives to document species turnover.

Both reviewers also highlight the need for at least some discussion on the causes of the changes in the faunal assemblages over time, and a comparison with other data sources, such as data collected in long-term ecological studies. Reviewer 1 provides some suggestions.

A: We now added a paragraph to the discussion (p. 27) comparing the results of long-term ecological studies to our findings.

The second reviewer also points out that a change in a country's territory may also limit the use of natural history collections (see discussion).

A: We fully agree with the reviewers' concern. We addressed this by removing geographic areas that were underrepresented at certain times from the analysis.

Sincerely,

Reviewer comments to Author:

Reviewer: 1

Comments to the Author(s)

RSOS-201983: Ewers-Saucedo, C. et al. Natural history collections reconstruct 200 years of faunal change

The paper presents the results of a very interesting approach to use natural history collection data to reconstruct faunal change. The paper reveals the importance of natural history collections as archives of biodiversity changes. The authors found trends in species changes especially for larger and dominant species similar to earlier studies for collection data.

The paper is well written and structured and the authors used comprehensive statistical analyses for the analysis of their data.

They discussed the pro and cons of using collection data, e.g. increasing sampling effort and correctness of species identification over time. Indeed, collection data can be used especially regarding presence/absence analysis of species as well as presence and increase of invasive species

In contrast, the attempt to use collection data for reconstructing changes in species abundance over time might be a problem, because collections data especially of larger epifauna species might only represent a subsample of the total catch.

I am missing in the discussion the comparison of the collection data with data from long-term ecological studies, such as the papers given below. Senckenberg am Meer is studying the epifauna communities in the south-eastern North Sea since 1998 on annual basis as part of Senckenberg's LTER North Sea Benthos Observatory. Similar studies are continued in the Dutch EEZ by colleagues from NIOZ and in the UK EEZ by colleagues from CEFAS. The comparison with the ecological data will provide the possibility to compare and confirm trends in abundance in collection data at least for the last two to three decades.

A: You are correct, this aspect has not been considered in the discussion. We described known changes in the fauna of North and Baltic Sea briefly in the introduction, citing some of the literature you suggest (p. 4). In the discussion, we added a paragraph detailing the results of long-term ecological studies, and compared them to our results (p. 27).

The discussion is in some parts repeating own results, which should be shortened and more focused to references. Being an ecologist, I am missing the discussion of reasons for the faunal change, which might not be the major focus of this paper, but will arise automatically while comparing trends with those from ecological studies.

A: We now mention the most probable reason, temperature change, at several points: during the introduction (p. 4) and discussion (pp. 26-27). We also explain in more detail why this study was unable to correlate changes in abundance to environmental variables (p. 27). In short, the museum collection data we used is not continuous enough, while long-term ecological studies have a much finer temporal resolution that allows for correlations. Museum collections can reconstruct older species trends, which are not covered by the "relatively" recent ecological studies.

I recommend the paper for publication after major revision.

Relevant papers to be considered:

Callaway, R., Alsvag, J., de Boois, I., Cotter, J., Ford, A., Hinz, H., Kröncke, I., Lancaster, J., Piet, G. Prince, P., Ehrich, S. (2002). Diversity and community structure of epibenthic invertebrates and fish in the North Sea. ICES J. Sea. Res. 59: 1199-1214.

Meyer, J., Kröncke, I., Bartholomä, A., Dippner, J.W., Schückel, U. (2016). Long-term changes in species composition of demersal fish and epifauna species in the Jade area (German Wadden Sea/ North Sea) since 1972. Estuar. Coast. Shelf Sci. 181: 284-293.

Neumann, H., Ehrich, S., Kröncke, I. (2008). Spatial variability of epifaunal communities in the North Sea in relation to sampling effort. Helgol. Mar. Res. 62: 215-225.

Neumann, H., Ehrich, S., Kröncke, I. (2008b). Temporal variability of an epibenthic community in the German Bight affected by cold winter and climate. Clim. Res. 37:241-251.

Neumann, H., Ehrich, S., Kröncke, I. (2009a). Variability of epifauna and temperature in the northern North Sea. *Mar. Biol.* 156:1817–1826.

Neumann, H., Reiss, H., Rakers, S., Ehrich, S., Kröncke, I. (2009b). Temporal variability of southern North Sea epifauna communities after the cold winter 1995/1996. *ICES J. Mar. Sci.* 66: 2233-2243.

Neumann, H., de Boois, I., Kröncke, I., Reiss, H. (2013). Climate change facilitated range expansion of the non-native Angular crab *Goneplax rhomboides* into the North Sea. *Mar. Ecol. Prog. Ser.* 484: 143–153.

Neumann, H., Diekmann, R., Emeis, K.-C., Kleeberg, U., Moll, A., Kröncke, I. (2017). Full-coverage spatial distribution of epibenthic communities in the south-eastern North Sea in relation to habitat characteristics and fishing effort. *Marine Environmental Research* 130: 1-11.

Rees, H. L., Pendle, M. A., Waldock, R., Limpenny, D. S., and Boyd, S. E. 1999. A comparison of benthic biodiversity in the North Sea, English Channel, and Celtic Seas. – *ICES Journal of Marine Science*, 56: 228–246

Reiss, H., Kröncke, I. (2004). Seasonal variability of epibenthic communities in different areas of the southern North Sea. *ICES J. Mar. Sci.* 61(6): 882-905.

Sonnewald, M., Türkay, M., 2012a. The megaepifauna of the Dogger Bank (North Sea): Species composition and faunal characteristics 1991-2008. *Helgoland Marine Research* 66(1), 63–75.

Sonnewald, M., Türkay, M., 2012b. Environmental influence on the bottom and near-bottom megafauna communities of the Dogger Bank: A long-term survey. *Helgoland Marine Research* 66(4), 503–511.

Sonnewald, M., Türkay, M., 2012c. Abundance analyses of mega-epibenthic species on the Dogger Bank (North Sea): Diurnal rhythms and short-term effects caused by repeated trawling, observed at a permanent station. *Journal of Sea Research* 73: 1-6.

Zühlke, R., Alsvag, J., de Boois, I., Cotter, J., Ehrich, S., Ford, A., Hinz, H., Jarre-Teichmann, A., Jennings, S., Kröncke, I., Lancaster, J., Piet, G., Prince, P. (2001). Epibenthic diversity in the North Sea. *Senckenbergiana marit.* 31(2): 269-281.

Etc.

A: We now include all bolded papers in our manuscript.

Reviewer: 2

Comments to the Author(s)

I have a great difficulty with this paper. This is very interesting and important contribution to science and shall be published. On the other hand it is a typical product of the „big data science” that can be produced about books in libraries, or minerals in geological museums. While this very paper is about living organisms. Yet, there is no natural history here – no information on the environmental context, climate, habitats or life traits of examined species.

A: This is indeed a valid concern. While we are unable to discuss the life history of all species, we are discussing the habitat preference of certain “outlier species”, e.g. *Palaemon serratus* (p. 21) or the unusual occurrence of shells of *Ostrea edulis* (p. 25). On page 22, we discuss the effect of cryptic morphology on species identification and misidentification. We added some more genus-specific information for declining genera on page 27.

Authors put great effort to present in most objective and bias free way the issue of museal collections, still it is a description of collection of certain class of objects from a politically designated area (German Baltic is treated differently prior to II WW and before). If some species were collected in 1900 near Königsberg in Eastern Prussia there is no reason to

ignore its presence in 1980 Kaliningrad in Soviet Union – while the only problem is that recent one is not in the German museum.

A: We fully agree with the reviewers' concern. We addressed this by removing geographic areas that were underrepresented at certain times from the analysis.

The analysis are very convincing, statistical treatment looks great – I am just curious what kind of results authors will get introducing the same method for the paleontological species – the same 200 years of history – will it show increase in abundance, distribution shifts and decline of some fossil species within 200 years time ?

A: We acknowledge your concern in this regard, and hoped to address this concern by comparing the museum collection data to other, independently generated data, such as long-term ecological surveys and other museum collections digitally available in GBIF.

Still this paper may indicate some important phenomena, that needs to be checked – habitat wise, climate and industrial stress considered etc...

I would recommend the paper to be published with a title changed to something like: „do changes in natural history collections reflect the changes in nature ? „
With some sentence in the intro – or conclusion that authors are not presenting the natural phenomenon – rather indicate that it may exist.

A: We amended the title, but did not formulate a question. When reading journal guidelines and advice on good titles, the general consensus appears to be that questions should be avoided. We changed the conclusions to meet your concerns.